# How Do Physiotherapists Explain Influencing Factors to Chronic Low Back Pain? A Qualitative Study Using a Fictive Case of Chronic Non-Specific Low Back Pain

**DOI:** 10.3390/ijerph20105828

**Published:** 2023-05-16

**Authors:** Rob Vanderstraeten, Antoine Fourré, Isaline Demeure, Christophe Demoulin, Jozef Michielsen, Sibyl Anthierens, Hilde Bastiaens, Nathalie Roussel

**Affiliations:** 1Department of Rehabilitation Sciences and Physiotherapy (MOVANT), Faculty of Medicine and Health Sciences, University of Antwerp, 2610 Antwerp, Belgium; rob.vanderstraeten@uantwerpen.be (R.V.);; 2Department of Neurosciences, Université de Mons, 7000 Mons, Belgium; 3Department of Sport and Rehabilitation Sciences, University of Liege, EVAREVA, 4000 Liege, Belgium; 4Faculty of Motricity Sciences, UCLouvain, 1348 Louvain-la-Neuve, Belgium; 5Anatomy and Research Centre (ASTARC), University Hospital of Antwerp, Antwerp Surgical Training, 2650 Antwerp, Belgium; 6Department of Family Medicine and Population Health (FAMPOP), Faculty of Medicine and Health Sciences, University of Antwerp, 2610 Antwerp, Belgium

**Keywords:** biopsychosocial, belief, chronic, behaviour, physiotherapy

## Abstract

Background: While pain is influenced by multiple factors including psychosocial factors, previous research has shown that physiotherapists still favour a biomedical approach. Purpose: To evaluate: (1) how physiotherapists explain the patient’s chronic non-specific low back pain (LBP); (2) whether physiotherapists use one or multiple influencing factors, and (3) whether these factors are framed in a biopsychosocial or biomedical approach. Materials and methods: This exploratory qualitative study uses a vignette depicting chronic non-specific LBP and employs a flexible framework analysis. Physiotherapists were asked to mention contributing factors to the pain based on this vignette. Five themes were predefined (“Beliefs”, “Previous experiences”, “Emotions”, “Patients behaviour”, “Contextual factors”) and explored. Results: Physiotherapists use very brief explanations when reporting contributing factors to chronic pain (median 13 words). Out of 670 physiotherapists, only 40% mentioned more than two different themes and 2/3rds did not see any link between the patients’ misbeliefs and pain. Only a quarter of the participants mentioned the patient’s worries about pain and movement, which is considered to be an important influencing factor. Conclusion: The lack of a multifactorial approach and the persistent biomedical beliefs suggest that it remains a challenge for physiotherapists to fully integrate the biopsychosocial framework into their management of chronic LBP.

## 1. Introduction

Low back pain (LBP) is the most frequent musculoskeletal disorder that people consult a general practitioner for [1]. It remains a huge economic burden for society, considering that 70–85% of all people will have LBP at some time in their life [2] and that 4–20% of them will develop chronic pain [3]. Moreover, a study evaluating years lived with a disability as a measure of disease burden reported that LBP was among the leading causes in 2017 [4].

The International Association for the Study of Pain defines pain as “an unpleasant sensory and emotional experience associated with, or resembling that associated with, actual or potential tissue damage” and highlights the multidimensional nature of pain [5]. Pain assessment and management is not straightforward as pain is not only defined by a possible noxious stimulus but is also influenced by previous experiences as well as personal and contextual factors [5]. These experiences and personal/contextual factors heavily influence patients’ pain. Indeed, pain beliefs and behaviours might explain the persistence of pain in some patients [6,7,8,9,10]. Hence the importance of addressing all these factors, including beliefs, emotions, and behaviour when managing a patient suffering from LBP.

Evidence-based guidelines for the management of LBP advise a biopsychosocial approach. The first step is to exclude a specific underlying cause of LBP [11,12,13]. The majority of people with LBP (>85%) are, however, diagnosed with non-specific LBP, which implies that it is hardly possible to identify the specific source of the nociception [14,15,16,17]. The second step is the identification of unhelpful beliefs, attitudes, emotions, behaviour, social factors, etc., of the patient (i.e., psychosocial factors often referred to as “Yellow Flags”), as these are indicative of poor outcomes [18,19,20,21]. However, research has shown that health care practitioners (HCPs) working in first-line care such as general practitioners and physiotherapists do not sufficiently assess these “Yellow flags” [22,23,24]. Moreover, many HCPs still have biomedical beliefs that pain can be reduced to a degeneration or anomaly of a body structure, independent of psychosocial factors, leading to biomedically-oriented advice restricting the patient in work or activities [25,26,27,28,29]. 

Several quantitative studies have been conducted to examine the knowledge, attitudes, and beliefs of HCPs regarding the management of patients with LBP, using questionnaires and vignettes [30,31,32,33,34,35,36,37,38,39,40,41]. These questionnaires and vignettes, often scored with a Likert scale, cannot fully represent how well HCPs integrate the multifactorial approach when managing pain. However, there is a lack of in-depth research exploring how physiotherapists actually explain the influence of multiple factors on the musculoskeletal pain of a patient with LBP. 

The objective of the current study is to: (1) evaluate how physiotherapists explain the patient’s chronic LBP; (2) observe whether physiotherapists use one or multiple factors to explain the patient’s pain; (3) to explore whether these factors are framed using a biopsychosocial or biomedical approach.

## 2. Materials and Methods

This exploratory qualitative study using a clinical vignette is part of the baseline assessment of a randomized clinical trial (RCT) registered on clinicaltrials.gov (NCT05284669). This trial uses an e-learning intervention to implement guideline-adherent care in first-line HCPs to enhance the knowledge, attitudes, and beliefs of HCPs towards a biopsychosocial approach in the management of patients with LBP. 

### 2.1. Sampling and Recruitment

Licensed Dutch- and French-speaking physiotherapists in Belgium and France were informed about the possibility to participate in an online study (RCT). Various strategies were used [42] to contact clinically active physiotherapists in Belgium and France. Invitations were shared in two languages (Dutch and French) in broad networks such as national associations (e.g., Axxon, Domus Medica, etc.), local networks of university departments and hospitals, registered physiotherapy associations, etc. Eligibility criteria were French-speaking or Dutch-speaking graduated physiotherapists working in Belgium or France. Exclusion criteria were physiotherapists not managing patients with low back pain or not being in possession of an internet-connected device.

### 2.2. Data Collection

Data collection of the RCT started in 27 August 2021 and ended in 1 February 2022. Participants were invited to fill in the online survey in their own language (Dutch or French) on their own device (e.g., computer, tablet, or smartphone) through the Qualtrics program (https://qualtrics.com, accessed 2 February 2022) after filling in their informed consent. For this exploratory qualitative study, relevant information was collected out of the online survey. Participants were asked to answer socio-demographic questions (age, estimation of new LBP patients treated/managed per month, gender, and years of experience). A fictive chronic non-specific LBP clinical vignette was specifically developed in French and Dutch for this study (see Appendix A). Participants received the instruction to read this vignette. Only the relevant information was included in the vignette. Participants were invited to answer the following open question: “In your opinion, what are the causes and/or contributing factors to this patient’s pain?”. Entries with blank answers to the open question of the clinical vignette were excluded.

### 2.3. Ethical Considerations

The ethical commission of the Antwerp University Hospital approved the study and written informed consent was obtained for all participants. The study was conducted in accordance with the General Data Protection Regulation. Data were automatically collected via the online survey instrument Qualtrics program often used by researchers with a strong confidential policy. 

### 2.4. Data Analysis

A mixed methods analysis of qualitative data was utilized using both a thematic framework approach and descriptive statistics [43,44,45]. In order to provide an answer to the research objectives and explore which factors physiotherapists take into account when evaluating a patient with chronic LBP, a framework was developed prior to analysis to evaluate the answers of the participants on the open question of the clinical vignette. Based on the current guidelines for the management of LBP, five relevant themes (factors) related to the vignette were predefined for a flexible deductive framework analysis (i.e., Beliefs (B), Previous experience with therapy (PE), Emotions (E), Patient behaviour (PB), and Contextual factors (CF)). For these five themes, the frequency of occurrence was counted in the answers. The International Association for the Study of Pain recognizes the need to consider pain as a personal experience that is influenced by many factors [5,46,47]. For this reason, when physiotherapists only used one or two themes to explain the cause and/or contributing factors to chronic LBP, this was considered as monofactorial. When the answer consisted of three or more themes it was scored as multifactorial. 

To explore whether the physiotherapists had a biopsychosocial vision of pain, for each theme in the framework, a description was provided, serving as guidance for the researchers (Table 1). This guidance allowed the researchers to classify the content of the answers in their respective themes as either “biopsychosocial quotes” (i.e., considering the clinical case, the answer of the participant was clearly in line with the evidence-based guidelines for the management of LBP) [12,13,18], or “biomedical quotes” (i.e., considering the clinical case, the answer as the main reason was not in line with biopsychosocial guidelines and/or implied a potential negative/harmful message and influence on the patient) [12,13,18] and these were, on their turn, counted for frequency of occurrence. Regarding the theme “Beliefs”, answers were further subdivided into three biopsychosocial subthemes (B1, B2, B3, Table 1) and three biomedical subthemes (B4, B5, B6, Table 1) with their own descriptions. 

Both the vignette and the framework with the predefined themes were pilot tested by a team of clinical and scientific experts (consisting of academics, general practitioners, orthopedic surgeons, and physiotherapists) on LBP controlling the content of the vignette and verifying the themes in the framework on their relevance and completeness towards LBP management. The answers of the Dutch and French physiotherapists were coded by native speaking researchers (R.V and I.D. and A.F. and I.D., respectively). All answers were coded by two independent researchers. To ensure correct interpretation of the description of the themes within the framework, the researchers individually analysed and compared answers of the first 100 participants. Differences in interpretations of the framework between researchers were discussed and a meeting was held with the expert team to ensure each researcher analysed and interpreted the framework in an identical way. 

During the analysis, an inductive analysis method was adopted when: (1) it was noted that some answers of the physiotherapists did not provide enough information to correctly interpret the reasoning behind the participant’s answer. These kinds of quotes were to be deemed as “neutral” during the analysis. Quotes were also deemed as “neutral” when a plausible explanation was given but the quote was based on information not present in the case. (2) When an answer was given that could not be classified in the aforementioned themes, they were categorized in the “Other” category. (3) A reflective analysis was undertaken to further explore the content answers (biopsychosocial-oriented approach versus biomedical orientation).

## 3. Results

### 3.1. Descriptive Data

In total, 670 participants enrolled in the study (female: 58%; male: 42%), with a median age of 30 (26–43) years. The majority of physiotherapists reported treating a maximum of ten new LBP patients per month. The socio-demographic results in Table 2 are similar between participants coming from the three regions except for the years of experience, which is lower in France. Of the participants in Wallonia and France, respectively, 2% and 15% failed to fill in at least one socio-demographic question.

### 3.2. How Do Physiotherapists Explain Contributing Factors to Pain

Physiotherapists used very brief explanations when reporting contributing factors to chronic pain with a median of 13 (7–24) words. When exploring which themes were most frequently mentioned (Figure 1), approximatively two thirds of the physiotherapists mentioned “Beliefs” or “Emotions” as contributing factors to the pain. However, a significant proportion of participants did not mention the “Patients behaviour”, “Contextual factors”, or “Previous experiences” in relation to pain.

### 3.3. Use of Multiple Factors When Explaining Contributing Factors to Pain 

Figure 2 depicts the number of different contributing factors used when explaining pain. Nearly two-thirds of the physiotherapists did not mention more than two different themes in their answers. Only 12% of the physiotherapists recognized four or all predefined themes in the case. There were no predefined themes present in the answers of seven physiotherapists and, therefore, they represent the “0” on the x-axis in Figure 2. 

### 3.4. Biopsychosocial vs. Biomedical Approach in Explaining the Pain

Table 3 details how physiotherapists quoted a (sub)theme, including examples of quotes. During the analysis of the answers, only a small minority of the quotes did not contain enough information to correctly interpret the presence of a biopsychosocial or biomedical approach, and these were considered neutral. 

Physiotherapists frequently mentioned the passing of the patient’s wife within either an emotional (grievance related to the loss) or contextual (e.g., related to the patient having to live alone) approach. In the vignette, only the death of the wife was mentioned, without any other information regarding emotions or context. As it is plausible that the passing away of the wife contributed to the patient’s pain, but as there was not enough information in the case related to this topic, these answers were considered as “neutral” (i.e., Emotions (E-n) and Contextual factors (CF-n)). 

When exploring biopsychosocial-oriented answers in Table 3, physiotherapists frequently mentioned the sedentary nature of the patient (PB1: 52%) or the plausible presence of the patient’s fear of movement (E1: 28%). Other themes were quoted in less than 20% of all participants, with ‘Previous experiences of therapy’ mentioned as the least frequent (PE1: 5%) contributing factor to the patient’s pain. It is, however, striking that many quotes reflect a strong biomedical orientation regarding this vignette describing a person with chronic non-specific LBP, such as beliefs attributing the pain to harmful biomechanical dysfunctions (B3: 23%) or to a specific underlying pathology (B6: 12%) (Table 3).

A small group of physiotherapists (9%) mentioned underlying or altered pain mechanisms as a contributing factor to the patient’s pain. These quotes were not included in the predefined themes (as no specific information related to pain mechanism was included in the vignette) and were therefore classified in the “Other” category.

### 3.5. Reflective Analysis

Compared to the other themes, the theme “Beliefs” showed a substantial number of biomedically-oriented quotes when explaining the main contributing factors to pain. When exploring the beliefs associated with pain, it appears that less than a quarter of the participants (23%) correctly identified the presence of at least one of the misbeliefs (Table 3, B1–B3) in relation to the patient’s pain present in the clinical case (Figure 3). One-third of the physiotherapists (33%) considered, among other factors, that biomechanical dysfunctions or pathology were the main reason for pain in this clinical case (Table 3, B4–B6), while one-third (36%) did not report anything at all about the pain beliefs in their answer (despite the presence of important misbeliefs in the case).

## 4. Discussion

Although physiotherapists in this study managed to recognize themes in line with the biopsychosocial framework [48,49], it is concerning that a considerable proportion of physiotherapists did not identify themes present in the clinical vignette as relevant factors contributing to the patient’s pain. Most physiotherapists still use a monofactorial approach to explain the patient’s pain and a third of the physiotherapists adopted a mainly biomedically-oriented pain explanation.

This is the first study demonstrating that physiotherapists are very succinct in their explanation of the contributing factors to pain. The extremely brief answers suggest that physiotherapists remain rather superficial and monofactorial when explaining the contributing factors to the patient’s pain. As this case relates to the story of a person suffering from chronic non-specific LBP, this finding merits further attention, as it does not match the patient’s strong need for clear, consistent, and personalised information regarding their condition [50]. Although this study used a written case, the relatively short answers allowed us to have a very clear vision of the physiotherapist’s main idea about contributing factors to pain. 

The observation that more than two-thirds of the physiotherapists have not linked any “patients’ misbeliefs” to pain is surprising, considering that patient’s misbeliefs are considered an important risk factor for developing/maintaining pain [7,10,51,52]. A third of the physiotherapists showed in their answers that they still have misbeliefs themselves related to the patient’s pain (e.g., linking anatomical structures related to the pain, etc.). This is in line with studies reporting guideline-inadherent behaviour of physiotherapists, such as persistent advice to protect the back, to rest, or to avoid movement [27,28,53]. Only a small number of physiotherapists evoked the misbeliefs in the vignette. This is a notable finding, as one of the major key messages of the clinical guidelines is to avoid recommendations such as rest or avoiding certain movements in non-specific LBP [13,17,18,54]. In fact, validated instruments use these misbeliefs to identify whether HCPs have inadherent behaviour to the guidelines [55,56,57]. It is well known that it is not possible to identify a specific nociceptive cause (e.g., anatomical/biomechanical dysfunctions) in people with non-specific LBP [15,16,17], especially when the pain is chronic. This aligns with the findings of some studies reporting that physiotherapists are still convinced that imaging is needed to identify the cause of LBP [27,53,56]. Not mentioning these misbeliefs or even reinforcing them can suggest a vulnerability of the spine, which is a notable risk factor for chronicity [58] or influencing patients towards a negative outcome [21]. 

The theme “Emotions” was the second most quoted theme, but only a quarter of the participants made the link with the patient’s worries about pain and movement, which is considered to be an important influencing factor [59,60]. Almost half of the participants reported the grief of the deceased wife as an influencing factor. However, nothing was stated in the vignette regarding grief or how the patient coped in the last year since the passing of the wife (see also methodological shortcoming). It remains plausible that the patient’s pain could have been influenced by this event. The loss of the wife was not intended to be a relevant factor when the case was developed; it is remarkable to see how many physiotherapists focused on this aspect of the vignette as a main factor contributing to pain while not mentioning the available relevant information, such as the fear of movement.

Only half of the physiotherapists in this study recognized the patient’s sedentary behaviour and thus the need for physical activity. This is considerably less than previously reported in other studies using a clinical vignette with Likert scale options, in which 62–80% of physiotherapists acknowledged the role of physical activity [27,53,56]. It seems likely that it is much easier to indicate important factors in LBP management using a Likert scale than it is to recognize the factors in a vignette through an open question, which was the case in the current study. The application of exercise therapy and its hypoalgesic effects on chronic LBP is a well-supported treatment modality [61]. However, it is a challenge to make chronic pain patients adhere to exercise therapy, as adaptations in central inhibition take time [62,63]. This makes it difficult for the patient to stay motivated or have confidence in the treatment strategy when pain reduction is their primary expectation, especially when the patient is fear avoidant to pain [62]. Additionally, patients stated the need to have supervision, follow-up support, and the reassurance of the HCPs in their exercise programs [62]. 

A significant proportion of participants did not mention the theme “Contextual factors” and almost none of the participants talked about the “Previous experiences”. The latter is remarkable as previous unsuccessful experience with therapy is associated with a less favourable outcome [64]. Finally, it is known that the physical activity levels of patients might be influenced by family members [65]. The fact that only a minority of the physiotherapists recognized the activity-limiting advice of the children is therefore surprising.

The results of this study confirm that physiotherapists still struggle to integrate a biopsychosocial framework into their management of non-specific LBP. Both quantitative and qualitative studies previously reported the concern that physiotherapists lack a biopsychosocial approach when managing pain [66,67] or only partially recognize cognitive, social, and psychosocial factors [68]. Nonetheless, physiotherapists seem to recognize the importance of the influence of biopsychosocial factors on non-specific LBP management [66,67,69]. This is similar to what was observed in this study, as some themes are recognized, albeit less frequently. This conclusion confirms previous reports highlighting the lack of training skills in the treatment and management of chronic pain [70]. This notion is supported by young and experienced physiotherapists reporting that they feel unprepared to fully incorporate the biopsychosocial framework into their management [66,67,68]. Although aware of the questionnaires, physiotherapists indicate that they would rather evaluate psychosocial factors through a social interaction process than by using validated instruments [66,67]. One could argue that this leaves physiotherapists vulnerable to their own biases and strengthens them in thinking in a repeating pattern.

It is imperative for practising physiotherapists, future research, and educational modules to emphasize the need for insights into the multifactorial nature of pain, particularly in the context of chronic pain. Despite the recognized efficacy of a biopsychosocial approach to chronic pain, its practical implementation still presents difficulties. Physiotherapists need to be more aware of the multifactorial aspect of pain including biological, psychological, and social influencing factors. Future educational modules should also focus more on communicative skills to recognize the perceptions, beliefs, and previous experiences of a patient and how to adequately manage them (e.g., motivational interviewing techniques).

### Strengths and Weaknesses

To the best of our knowledge, this study is the first to use exploratory research on how physiotherapists identify and incorporate several factors when explaining contributing factors to chronic LBP. The large sample size with a representation of different ages and work experiences is certainly a strength of the study. However, the results of this study should be seen in the light of some methodological considerations. Despite the fact that national and registered local organizations were contacted to spread the information about the study, recruitment bias cannot be excluded. Because of the general protection data regulation (GDPR), it was not possible to obtain access to the full mailing list of the members of the organisation to explore reasons for non-participation. Since we awarded physiotherapists completing the full study with accreditation points, it is possible that some participants were only interested in accreditation points. Secondly, this study was conducted in Belgium and France. Since pain is influenced by a variety of factors (including cultural and socioeconomic factors) [16], it is uncertain if these results can be properly transferred to other countries, especially given that the educational aspect of the profession may vary between different countries across Europe. Although it was not the aim of the study to compare the results of physiotherapists with different educational backgrounds, these insights might be interesting for future studies. Thirdly, due to the short answers participants gave to the open question, some were therefore classified in the neutral category. Fourth, a methodological shortcoming of the vignette is that we did not provide much information about how the patient was affected by the death of his wife. This lack of information made it difficult to accurately interpret these quotes, causing a proportion of the answers to be classified as neutral. The information included in the vignette was rather succinct, as is the case in other vignettes (e.g., Rainville [71] and Bombardier [72]) used in many previous publications [28,73,74,75,76]. In research where vignettes are used, only the relevant information to participants is used. Therefore, our vignette did contain important information related to the multifactorial nature of LBP, including the absence of red flags. Based on the given information, participants should have recognized that this vignette described a patient with non-specific LBP, where the pain is influenced by different factors, including psychosocial factors (known to be associated with persistent pain). This approach is in line with all guidelines regarding the management of people with LBP. 

## 5. Conclusions

Most physiotherapists still use a very succinct, monofactorial approach when explaining contributing factors to chronic LBP, with persistent biomedical beliefs and insufficient attention towards fear of movement. It seems that it remains a challenge for physiotherapists to fully integrate the biopsychosocial framework into their management of chronic non-specific LBP. Future research should incorporate a multifactorial approach when exploring a biopsychosocial approach in the management of non-specific LBP.

## Figures and Tables

**Figure 1 ijerph-20-05828-f001:**
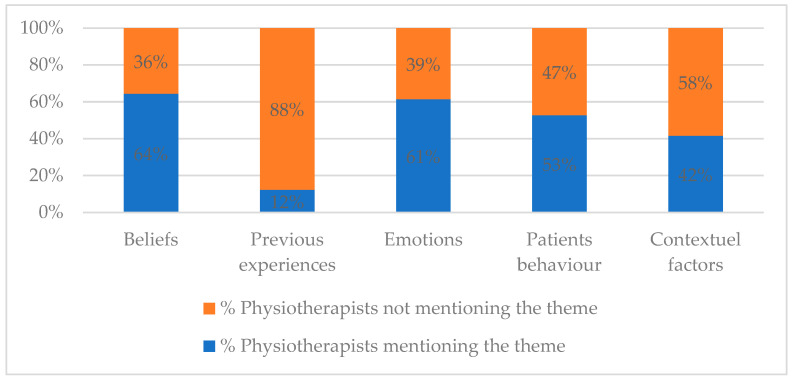
Frequency of themes.

**Figure 2 ijerph-20-05828-f002:**
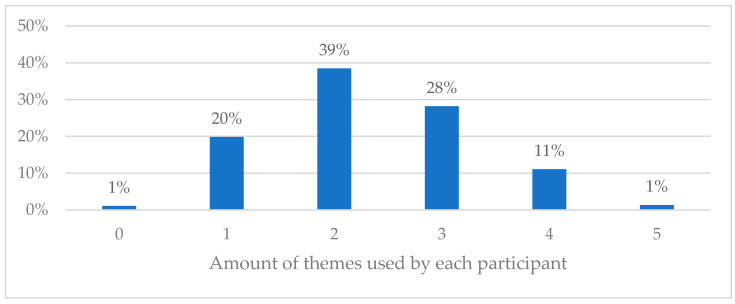
Use of multiple factors when explaining contributing factors to pain.

**Figure 3 ijerph-20-05828-f003:**
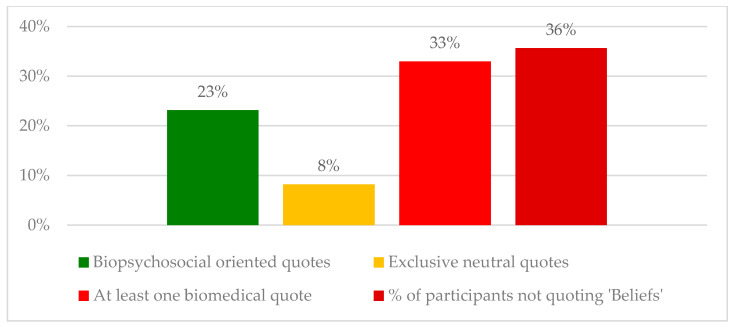
Reflective analysis of the theme Beliefs.

**Table 1 ijerph-20-05828-t001:** Relevant themes and their description, contributing to chronic pain in the clinical vignette.

Themes	Approach	Code	Description
Beliefs	Biopsychosocial	B1	Misbeliefs of the patient that pain is linked to tissue damage or to a biomechanical cause
		B2	Misbeliefs of the patient that rest/avoiding movement will lead to a decrease in pain
		B3	Misbeliefs that moving in a very specific way is necessary to decrease the pain
	Biomedical	B4	Attributing the cause of pain to bad movements or postures, etc.
		B5	Linking pain to ageing
		B6	Describing a specific pathology or impairment in anatomical structure as cause of the pain
Previous experiences	Biopsychosocial	PE1	The lack of success is explained by previous treatments that were too biomechanically focussed
	Biomedical	PE2	The lack of success is explained by the fact that the previous treatment was not well enough executed in a biomechanical framework
Emotions	Biopsychosocial	E1	The fact that the patient worries about the pain and about movement might contribute to the pain
	Biomedical	E2	Depression or a mentally fragile situation is the cause of pain
Patient’s behaviour	Biopsychosocial	PB1	The sedentary aspect or the avoidance behaviour of the patient might contribute to the pain
	Biomedical	PB2	The patient suffers from pain because he did not do his exercises well enough or did not comply enough
Contextual factors	Biopsychosocial	CF1	The unhelpful influence of the family might be related to the pain
	Biomedical	CF2	The patient is not able to adapt to his changed environment

B: Beliefs; PE: Previous experiences; E: Emotions; PB: Patient behaviour; CF: Contextual factors.

**Table 2 ijerph-20-05828-t002:** Socio-demographic information.

	n (%)			
	Flanders	Wallonia	France	Total
Population	308 (100%)	189 (100%)	173 (100%)	670 (100%)
Gender				
Female	201 (65%)	106 (56%)	92 (53%)	399 (60%)
Male	107 (35 %)	83 (44%)	81 (47%)	271 (40%)
Estimation new LBP patients/month				
1–5	72 (23%)	71 (38%)	28 (16%)	171 (26%)
5–10	106 (34%)	75 (40%)	68 (39%)	249 (37%)
10–15	69 (22%)	27 (14%)	29 (17%)	125 (19%)
15–20	24 (8%)	9 (5%)	15 (9%)	48 (7%)
>20	37 (12%)	4 (2%)	7 (4%)	48 (7%)
	Median (Q1–Q3)			
	Flanders	Wallonia	France	Total
Age (years)	34 (27–47)	30 (26–40)	27 (25–33)	30 (26–43)
Work experience (years)	10 (4–25)	7 (3–18)	5 (2–10)	7 (3–21)

n: amount; Q: quartile; LBP: low back pain.

**Table 3 ijerph-20-05828-t003:** Content and proportions of quoted themes.

Theme	Differentiation	Code	N%	Example of Quotes
Beliefs	Biopsychosocial approach	B1	12	“*Misinterpretation that pain equals damage*”
		B2	16	“*The belief that rest is necessary*”
		B3	14	“*Advise to avoid certain movements*”
	Neutral	B-n	15	“*His insight of pain*”
	Biomedical approach	B4	23	“*Wrong posture of the back*”, “*Weak muscle strength*”
		B5	4	“*Age*”
		B6	12	“*Osteoarthritis with possible disc problem*”
Previous experience	Biopsychosocial approach	PE1	5	“*Conflicting information between professionals*”
	Neutral	PE-n	6	“*No explanation about his back pain*”
	Biomedical approach	PE2	3	“*Incorrect exercises*”
Emotions	Biopsychosocial approach	E1	28	“*Fear of movement*”, “*Fear of pain*”
	Neutral	E-n	46	“*Grief due to death of wife*”
	Biomedical approach	E2	4	“*Mental fragility*”
Patients behaviour	Biopsychosocial approach	PB1	52	“*Lack of physical activity*”, “*Lack of exercise*”
	Neutral	PB-n	1	“*Lifestyle*”
	Biomedical approach	PB2	0.1	“*Poor exercise performance by the patient*”
Contextual factor	Biopsychosocial approach	CF1	18	“*Overprotection of the children*”
	Neutral	CF-n	32	“*Alone without his partner*”
	Biomedical approach	CF2	1	“*Change of activities in connection with the loss of his wife, adaptation necessary*”

N%: Percentage of the total population having a quote categorised in a (sub)theme, B: Beliefs, PE: Previous experience with therapy, E: Emotions, PB: Patient behaviour, CF: Contextual factors, -n: neutral.

## Data Availability

The data presented in this study are available on request from the corresponding author. The data are not publicly available due to the written consent of participants for staying anonymous and GPDR.

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
