# Peer review of "How Do Physiotherapists Explain Influencing Factors to Chronic Low Back Pain? A Qualitative Study Using a Fictive Case of Chronic Non-Specific Low Back Pain"

_ijerph, 2023, doi:10.3390/ijerph20105828_

Round 1

Reviewer 1 Report

Comments:

Thank you for letting me read your interesting, and very clearly written, manuscript!

I have a few questions that left me in some worry regarding the transferrability of your presented results: They are:

*This is mostly a quantitative study, looking at the recruiting of the large study population and the presentation of data mostly by large numbers.  Some qualitative data is presented. So your present at the most a mixed method study.
* Why did you choose a self-recruitement study population and not a strategic or similar recruitement method? You don´t discuss in detail this option in the method/discussion section.
*The vignette case is very brief and lacks observational data. Please, comment what that might mean for the responders, and the resultant answers.
* Do you think your presented data/findings are possible to transfer to other lets say European countries?

Author Response

Dear Reviewer,

Please find our responses in the attachements.

Kind Regards

Reviewer 2 Report

Dear Author

This study is the first to identify physiotherapy problems for LBP from a biopsychosocial perspective and is of high clinical value.

On the other hand, there are a number of points that need to be revised, see below.

Major points

l   I believe that a biopsychosocial framework is necessary within physiotherapy for LBP, but this is expected to be influenced by intellectual aspects such as the final education of the physiotherapist. The author's opinion on this point should be discussed or described as a limitation of the study.

l   The results of this study confirm that physiotherapists still struggle to integrate a biopsychosocial framework in the management of non-specific LBP, but the reader would like to know what improvements can be made to address this problem. Please add the author's opinion to the discussion.

Mainor points

・L40:Low back pain (LBP) is the most frequent musculoskeletal condition

Low back pain (LBP) is the most frequent musculoskeletal disorder

・L205-207, L214-216 and 223:Please include this information in the manuscript. It makes the manuscript more difficult to read.

Author Response

Dear Reviewer,

Please find our responses in attachment.

Kind Regards

Round 2

Reviewer 1 Report

Dear authors,

You have done a great work in improving your first version of the manuscript.